# The Potential of Microgranular Fertilizers to Reduce Nutrient Surpluses When Growing Maize (*Zea mays*) in Regions with High Livestock Farming Intensity

**DOI:** 10.3390/life14010081

**Published:** 2024-01-03

**Authors:** Frank Eulenstein, Julian Ahlborn, Matthias Thielicke

**Affiliations:** 1Department Sustainable Grassland Systems, Leibniz Centre for Agricultural Landscape Research (ZALF), Gutshof 7, 14641 Paulinenaue, Germany; feulenstein@zalf.de; 2Senckenberg Museum of Natural History Görlitz, Botany Division, Am Museum 1, 02806 Görlitz, Germany; julian.ahlborn@senckenberg.de

**Keywords:** microgranules, eutrophication, maize, phosphorus, DAP

## Abstract

This contribution provides the first agroeconomic account of the application of a mineral microgranular fertilizer and an organomineral microgranular fertilizer directly beneath the corn in comparison to a common mineral band fertilizer in temperate climate regions. The focus of the study is on the reduction in phosphorus inputs while maintaining the yield of maize plants (*Zea mays*). The study used a three-year field trial to tabulate dry matter yields using the two phosphorus-reduced microgranular fertilizers, as well as a standard diammonium phosphate (DAP) fertilization method. The application of the organomineral microgranular fertilizer resulted in dry matter yields that were 15% higher (2.8 Mg per hectare) than the DAP variant, while higher yields using the mineral microgranular fertilizer only occurred in a single year. The higher yield of the organomineral microgranular fertilizer and the lower phosphorus amounts as a result of using that product resulted in a moderate phosphorus excess of 2.7 kg P ha^−1^, while DAP fertilization resulted in a surplus of 25.5 kg per hectare. The phosphorus balance on the plots fertilized with the mineral microgranular fertilizer followed a pattern similar to that of the organomineral microgranular fertilizer. We conclude that both microgranular fertilizers, applied directly beneath the corn, provide an adequate alternative to widespread DAP fertilization as a fertilizer band in maize cultivation on fertile soils.

## 1. Introduction

The intensification of agriculture and the corresponding excessive phosphorus inputs have led to the eutrophication of ground and surface water systems across Europe [1,2]. It is undisputed that the phosphate rock resources for conventional fertilizer production are finite and must be used sustainably, although the extent of those resources is a subject of controversy in the literature [3,4]. Political restrictions designed to reduce the amount of excess nitrogen and phosphorus in agricultural areas have also taken on a new quality in Europe. In Germany, for instance, farmers have been compelled to invest in expensive, resource-consuming transport of their farm fertilizers, which are predominantly in liquid form, to areas with lower densities of livestock units [5,6]. To maintain the high yields necessary to compete in the globalized market, farm fertilizer has partly been replaced with mineral fertilizers, such as diammonium phosphate (DAP). The search for tools to address these challenges is driving a rapidly growing market of new seeding techniques and alternative fertilizers in microgranular form, designed to increase the precision of nutrient and seed placement. Microgranular fertilizers are introduced to the soil along with the seeds at a distance of a maximum of a few centimeters. This method contrasts with DAP and other fertilizers that are applied as a band about 7–12 cm from the seed. The small distance between a microgranular fertilizer and the seed requires smaller amounts per plant of both fertilizer and nutrients, especially phosphorus (P), to be used, and requires the components of the fertilizer itself to have a lower salt index [7]. The granules can have a mineral, organomineral, or organic origin and are smaller than 2 mm in diameter; their dispersal prevents long-term osmotic gradients. Cheap by-products from food processing, such as oil cake and bone meal, are usually used as organic compounds for manufacturing microgranular fertilizers. These by-products are regionally available and thus avoid the additional import of finite resources, such as rock phosphate, for mineral fertilizer production. Furthermore, previous research has shown that oil cake and bone meal act as biostimulants on soil bacteria [8,9], which may increase the positive effect of fertilizer application on the rhizosphere. Recently, two studies in Poland have also made progress in evaluating the effect of microgranular fertilizer in potting and field experiments under various climatic conditions [10,11]. However, it is still unclear how microgranular mineral and organomineral fertilizers directly compare to conventional fertilizing systems, especially in the context of modern, highly efficient agricultural practices in regions with lower soil temperatures during the early development of maize. DAP and microgranular fertilizers are considered starter fertilizers, providing readily available nutrients during maize’s sensitive early stages of plant development. In comprehensive studies carried out in the U.S. Corn Belt, the additional application of starter fertilizer placed in the furrow (i.e., the same placement as for microgranular fertilizer) failed to result in higher yields. This was the case irrespective of the soil’s P-contents (ranging from “low” to “over-fertilized”) or the management of the rye cover crop [12,13]. Only no-till sites showed positive short-term effects from the application of starter fertilizer [12]. This latter finding might imply that any significant effects of starter fertilizer are more likely to be achieved in colder soils, such as those found in central and northern Europe. A recent study from northern Italy appears to be an example of this. There, starter fertilization resulted in stronger crop performance, including the decisive factor of yield [14].

In general, only a few studies have compared the typical placement for microgranular fertilizer—i.e., in the furrow, directly on or a few centimeters distant from the seed—with fertilizer band placement [15,16,17,18]. Even as early as the 1950s, scientists identified band-fertilizer application to be the most effective method for placing nutrients for maize [19,20], especially in combination with broadcast fertilization of P; a more recent study has confirmed this [21]. However, the new techniques of precision agriculture enable the time-efficient and exact placement of small quantities of fertilizer.

The purpose of this investigation was to study the effects of two different types of microgranular fertilizer—one of which had organic and mineral compounds with a small application rate of 25 kg ha^−1^ and the other only mineral compounds—and to compare both to DAP band fertilization. Dry matter yields, corncob ratio, and nutrient balances reflect the economic potential of the two fertilizing systems under study. This entailed performing a preliminary agroeconomic calculation of the standard DAP fertilizer and the most suitable microgranular fertilizer variant for the study site.

## 2. Materials and Methods

### 2.1. The Study Area and Experimental Setup

Experiments were carried out as field trials with four fertilizer variants, each repeated four times, on plots measuring 13 meters in length and 3 meters in width near Wadersloh, in western Germany (51.716995 N, 8.241482 E). The region is classified as having a European Atlantic climate (Cfb) as defined by Köppen and Geiger (1930) [22], characterized by mild winters and moderate summer temperatures. The average precipitation per year for Wadersloh is 673 mm, and the average annual temperature is 10.4 °C. Around 53% of the annual rainfall occurs during the maize vegetation period from April to September. Loamy sand with a humus content of 2.5%, an effective field capacity of 9% (volumetric), and a pH of 5.7 is present on the study site. A calcium-acetate-lactate extraction (CAL-extract) [23] of the soil revealed moderate amounts of phosphorus (6.5 mg 100 g^−1^), potassium (6.6 mg 100 g^−1^), and magnesium (3 mg 100 g^−1^).

The site has been used for maize cultivation for around 10 years without crop rotation and treated with the following plant protectants: Gardo Gold^®^ (metolachlor and terbuthylazine) applied at the end of April at a rate of two liters per hectare and at the beginning of June at a rate of one liter combined with 0.3 liters of Buctril^®^ (bromoxynil), 0.9 liters of MeisTer^®^ (foramsulfuron, thiencarbazone, iodosulfuron, and cyprosulfamide), one liter of manganese nitrate, and 0.2 liters of Zeavit (oat-derived phytochemicals) per hectare. Regular tillage operations performed before sowing in early spring (March/April) involved ploughing at a depth of 25 cm and, two times, grubbing to a depth of 6 and 10 cm, respectively. Following this representative long-term fertilization system of the study site, we set up pre-treatment and control using pig slurry (20 m^3^ ha^−1^) containing 8.6 kg of total N, 2 kg of P, 4.9 kg of K, 2.4 kg of S, and 10.8 kg of magnesium per m^3^. 

The “Farmpilot” maize cultivar was sown at a density of 85,000 seeds per hectare using an AMAZONE EDX 6000-2C precision air seeder (Amazonen-Werke H. Dreyer SE & Co. KG, Am Amazonenwerk 9-13, 49205 Hasbergen, Germany), a single corn seeder system, and the microgranular fertilizers Startec (De Ceuster Meststoffen *NV* (DCM), Bannerlaan 79, 2280 Grobbendonk, Belgium) and Wolf Trax Nu-Trax P+^®^ (WolfNP) (MTD Products Inc., 5903 Grafton Road, Valley City, OH 44280, USA) were both applied a few centimeters beneath the corn kernel. The DAP fertilizer was applied in a band 12 cm below the soil surface, at a rate of 100 kg ha^−1^. It contained 18% total N, all in the form of NH_4_-N, and 20% P. Startec can be classified as an organomineral fertilizer, of which 80% (of the original substance by weight) is made up of the organic industrial by-products, oil cake and deglued bonemeal, and the mineral components, ammonium phosphate, ammonium sulphate, EDTA-chelated Fe, Mn, Zn, zinc sulphate, and zinc oxide. The nutrient composition of Startec is 7.5% N, 9.6% P, 3.3% K, 4% S, 0.5% Fe, 0.5% Mn, and 1.5% Zn. For this study, Startec was applied at a rate of 25 kg ha^−1^. WolfNP is classified as a mineral microgranular fertilizer and contains ammonium sulphate, kieserite (magnesium sulphate as a monohydrate), and Nu-Trax P+. This final component is a fine-grained water-soluble powder that serves as a coating for the product. WolfNP was applied at a rate of 100 kg ha^−1^, and it has the following mineral composition: 11% N, 0.2% P, 8.8% S, 7.5% Mg, 0.4% Mn, and 0.1% Zn.

Application rates and nutrient inputs per hectare for each type of fertilizer are summarized in Table 1. Differences in nutrient inputs reflect the application rates prescribed by the manufacturer of the respective fertilizer and their use in agricultural practice. 

### 2.2. Data Sampling and Statistical Analysis 

Hand harvest was performed by randomly removing 20 plants per plot, 15 cm above the ground. The corncob and the remaining plant were weighed and shredded separately using an AL-KO Master 32–40 garden shredder. The shredded material of the corncob and the remaining plant was used to determine the content of dry matter, raw protein, phosphorus, and potassium for each of the four repetitions of a variant. We then used the latter data to calculate the effect of harvesting on the year-specific removal of N, P, and K.

We used the “R” software package to verify a normal distribution of the data. Significant differences between the variants were tested using an analysis of variance (ANOVA) test, followed by Tukey’s post hoc test for multiple comparisons of the dry matter yield (=yield) and corncob ratio (the proportion of dried corncobs relative to the total plant dry matter).

Fresh matter and calculated silage yield were excluded from statistical comparisons. All statistical analyses were performed using R software version 4.1.3. For data selection, we used the dplyr package [24]. The visualization in R was conducted using the ggplot2 package [25] and Microsoft Excel. 

For the agroeconomic calculation (Table 2), we compared the silage yields of the two fertilizer variants, DAP and Startec, to their costs. We calculated silage yields by assuming a 12% silage loss of the fresh matter yield. The silage price was set according to the lowest price boundary of the last seven years (own database). The costs include purchase prices as well as the required quantity of liquid manure to be removed. The transport prices for the liquid manure were determined on the basis of a separate survey from 2021 of 14 farms in northwest Germany, which resulted in an average cost of €4.87 per m^3^ slurry. This average value was set as a constant and multiplied by the surplus of slurry (in m^3^) to be removed. The necessary volume of slurry a farmer has to dispose of is based on the nutrient analyses of the pig slurry used in the experiment and the legally defined limits for the nutrient load that farmers are permitted to introduce to their fields. The regulatory limits used in the sample calculation are based on the directly enforceable law of the German Fertilizer Directive (DüV, Article 97, Ordinance of 10 August 2021, BGB p. 3436), Germany’s national implementation of the European Union Water Framework Directive (91/676/EEC, 2000/60/EC). To briefly note the relevant parts of the directives for the following estimates, an excess of 4.36 kg P per hectare per year (over a six-year average) in the fertilizing balance is permitted if the extractable soil phosphorus is ≤8.7 mg per 100 g of soil following the extraction procedure described by Schüller (1969) [23] using the calcium-acetate-lactate method (CAL extract), or ≤10.9 mg per 100 g of soil after double lactate (DL) extraction [26], or ≤3.6 mg using the electro-ultrafiltration method [27]. No excess is permitted if it is possible to extract more phosphorus by following the above methods.

## 3. Results

### 3.1. The Effect of Microgranular Fertilizer on Dry Matter Yield and Nutrient Balances

Figure 1 depicts all results corresponding to the yields from different fertilizer plots. Comparing the yields from the control plots and the DAP-fertilized plots revealed no significant differences. Significant differences in yield between the control and fertilizer variants only occurred in 2016 for WolfNP and Startec (*p* < 0.001). The average dry matter yield (whole plant) per hectare obtained using the organomineral microgranular fertilizer (Startec) during the study was 2.9 Mg (15%) higher than the yield obtained using DAP. This result was statistically significant (*p* = 0.028). While the yields from Startec plots were only 10% higher than those from DAP plots in 2015 (and thus not statistically significant), the difference in 2016 was 3.8 Mg and thus highly significant (*p* < 0.001) and was 3.2 Mg higher in 2017. The P balance of Startec-fertilized plots was significantly (*p* < 0.001) lower each year than that of DAP-fertilized plots (Figure 2).

The average overall yields of the three-year study (and the annual yields for 2015 and 2017) showed no statistically relevant differences between the mineral microgranular fertilizer (WolfNP) and DAP. In 2016, the application of WolfNP resulted in higher yields compared to DAP fertilization. The difference of 3.3 Mg was statistically significant (*p* < 0.001). The P balance of the WolfNP plots compared to the DAP plots exhibited the same pattern as that described above for Startec in each year (*p* < 0.0001). However, the phosphorus excess following the application of WolfNP was 7.7 kg P ha^−1^. Statistically relevant differences in the N balance only occurred in 2016 in comparison to Startec (*p* = 0.027) and DAP (*p* < 0.001) (Figure 3). The average excess nitrogen on plots treated with WolfNP was 5.7 kg N ha^−1^ over the three-year study. The proportion of corncobs relative to the total mass of the plant varied over the three years, ranging from 57% in 2017 to 65% in 2016. Significant differences between the treatments only occurred in 2015, when the corncob ratio of WolfNP-fertilized plants was higher than the control (*p* = 0.023).

### 3.2. Economic Analysis of DAP and the Startec Organomineral Microgranular Fertilizing System

All results corresponding to the economic comparison between DAP and Startec are presented in Table 2. The hectare-specific costs for Startec were €60 and thus higher than for DAP (€40). The P excess on Startec plots was below the tolerance range and consequently led to no slurry reduction. The necessary slurry reduction on DAP plots would be 8.6 m^3^, which results in slurry disposal costs of €41.11. The resulting fresh matter yields of silage were 53.67 Mg from DAP plots and 60.92 Mg from Startec plots. The 7.25 Mg higher silage yield from Startec plots resulted in a higher return of €253.75 per hectare compared to DAP. The agroeconomic comparison of the two fertilizer variants DAP and Startec resulted in a lower cost–benefit ratio for the latter fertilizer.

## 4. Discussion

### 4.1. Comparing Yield and Nutrient Balances with DAP and Microgranule Fertilization

When comparing the four different fertilizing systems (DAP, WolfNP, Startec, and the control), it is clear that the amounts of phosphorus and nitrogen per unit of area vary (Table 1). Reducing macroelements, especially phosphorus, has been found to limit yields in plant growth [28]; however, this does not seem to have been the decisive factor in the fertile soil of our study site. The fertilizer treatment with the highest P input (DAP) resulted in similar yields as that of the variant with the lowest P influx (the control treatment (slurry only), without any additional fertilizer) (Table 1). Previous studies have similarly shown that differences in P inputs and fertilizer types do not automatically result in different levels of P forms available to plants within the soil P pool and thus yields, depending on the crop [29,30]. Furthermore, the soil of the study site was already moderately rich in P, which may have led to the fertilizer having a smaller effect. In a comprehensive review of several field studies, Quinn et al. (2020) [18] described a decreasing effect of band fertilization if the soil P is high, which does not necessarily mean that yield increases due to fertilizer application only occur in P-deficient conditions. The only significant differences between WolfNP and Startec, on the one hand, and the control, on the other, occurred in 2016, when both fertilizers led to higher yields than the slurry-only treatment. It is crucial to point out that data that are statistically insignificant in scientific considerations cannot in each case be taken into account in the same theoretical way regarding their relevance in practice. The Startec organomineral fertilizer resulted in 2.7 Mg more harvested dry matter per year compared to the control plots, which were only pre-treated with pig slurry. This trend would have a major economic effect on agriculture, although it is not statistically significant for the present study. A clear pattern with economic relevance can furthermore be found in the arithmetic mean yield gained using Startec, which was higher every year than all the other average yields from other variants. In temperate regions, the application of DAP is a widespread agricultural practice; the use of microgranular fertilizers is relatively rare, although the market share of this has grown in the last decade. The average increased yield with Startec (2.9 Mg ha^−1^) compared to DAP plots was statistically significant. This confirms that Startec is a more suitable fertilizer under the given conditions. However, when considering each year separately, it was found that the yield from plots fertilized with Startec was only in the significant range in 2016. Interestingly, in 2015, the difference between the yields generated by Startec and DAP variants was in the insignificant range of 10% (*p* = 0.17); this year also witnessed the lowest total difference between the yields of the other fertilizer variants at any point during the study time. In 2015, the early vegetation period of maize was comparatively dry, with 62 mm of precipitation in total over the months of May and June. That was approximately half the rainfall of the same period in 2016 and 2017 during the same period. Thus, water might be the key factor rather than the different nutrient influxes or other modes of action in terms of biostimulation resulting from Startec’s organic compounds, which we discuss at the end of this section. Previous studies have asserted that crucial changes in weather exceed the nominal effect of a given fertilizing system [17,31].

The proportion of corncobs relative to the total mass of the plant varied over the three years, from 57% in 2017 to 65% in 2016. Among the fertilizer variants, significant differences only occurred in 2015, when the corncob ratio of WolfNP-fertilized plants was higher than the control. With one exception of WolfNP in 2016, each fertilizer variant resulted in 2.6% higher corncob ratios compared to the control on average. Although this is not a statistically significant trend, fertilization (in addition to the slurry treatment) has an impact on the corncob ratio. Complementing the differences in yield, the disparities in nutrient balances are higher. This is due to the different compositions of the different fertilizers. The P balance of plots fertilized with Startec was +2.7 kg P per hectare, while DAP fertilization resulted in +25.5 kg per hectare. The average P balances over the three years and those from the individual years clearly and significantly differed. In summary, Startec was an adequate alternative to DAP fertilization in terms of yield, as well as of N and P balances for fertile fields, if they have comparable conditions to those of the study site. The only statistical differences in terms of the N balances of plots fertilized with Startec and DAP were present in 2016, although the N balances obtained using Startec resulted in an amount of −35.8 kg N per hectare, while the use of DAP resulted in an increase of 30.6 kg N per hectare. Differences in this range are highly relevant for agronomical practices and for the environment. A previous study found similar results comparing the yields and nutrient balances of Startec and DAP in fertile loamy soil in northern Germany [32].

Over the three years, the dry matter yield from the mineral microgranular fertilizer variant (WolfNP) was nearly the same as that of DAP (4% higher than DAP). 2016 possibly witnessed the best possible conditions for WolfNP, because the dry matter yield was 3.3 Mg per hectare higher than that resulting from DAP plots. The P balance exhibited similar patterns, with highly significant differences as described above for Startec, in WolfNP compared to DAP-fertilized plots. The phosphorus excess with WolfNP was 7.7 kg P ha^−1^ and thus higher, albeit not statistically significant, than the P balance of Startec plots. This discrepancy can be explained by the higher dry matter yield from Startec plots. Differences between the three N balances of DAP, WolfNP, and Startec plots were of statistical significance in 2016 and can be ascribed to the aforementioned differences in yield. The N influx of the fertilizer variants themselves differs slightly due to the high N influx by the slurry pre-treatment (Table 1). The average excess of nitrogen on plots treated with WolfNP was 5.7 kg N per hectare over the three-year study, which is close to neutral and thus serves as a goal for future sustainable agriculture. However, the application of the organomineral microgranular fertilizer (Startec) resulted in higher yields and a lower nutrient surplus than the microgranules of mineral origin used in WolfNP. This may be because they have a more beneficial nutrient composition for maize cultivation on loamy sand under the given climatic conditions. Furthermore, the organic compounds in Startec exert additional biostimulative effects. Startec’s organic compounds, such as oil cake and bone meal, act to a certain extent on microbial activity. Oil cakes are used to raise microbial metabolism in the bioremediation of soils [33], as well as other biotechnological applications [9]. Bone meal is known to increase mineralization dynamics and thus extractable macronutrients in soils [8] and to act as a biostimulant for bacteria [34]. Contrary to the well-described introduction of defined microbial inoculants in agricultural soils [35,36,37,38,39,40], recent studies have paid attention to the recovery of indigenous soil microbial biomass and diversity by replacing some of the inorganic fertilizers with organic fertilizers [41,42,43]. Startec’s organic compounds might support the indigenous microbiome in a manner similar to the yield-increasing effects, analogous to biogas residues, as reported by Zaho et al. [43], or to dairy cattle manure and rapeseed cake, as documented in Wang et al. [42]. Differences in yields in our study due to the varying micronutrient inputs of the different fertilizers were negligible and overlapped by the nutrient inputs of the slurry pre-treatment.

### 4.2. Economic Consideration of DAP and the Startec Organomineral Microgranular Fertilizing System

In agricultural practice, fertilizing systems will only succeed if they are economically viable. A brief comparative agroeconomic calculation for the two fertilizer variants of DAP and Startec is summarized in Table 2. The regulatory limits used in the sample calculation were based on the directly enforceable law of the German Fertilizer Directive (DüV, Article 97, Ordinance of 10 August 2021, BGB p. 3436). To maintain high yields, farmers usually reduce organic fertilization but not mineral fertilization. Excess farm fertilizer is sent to farms with better nutrient balances, such as those in areas with low densities of livestock units. The transport of slurry, which usually has a water content of over 90%, consumes a great deal of resources and is not well accepted among the general public. According to an own previous survey, farmers expect the cost of disposing of this slurry to be about €4.87 per m^3^. However, rates of €10 per m^3^ or more also occur in German agriculture. We calculated a relatively high 12% for fresh-matter loss during ensilage and a relatively low price for silage to avoid any overestimations for the calculated example. The nutrients in slurry are also 25–40% higher in the case of N and 12–18% higher for P compared to average values for pig slurry [44]. Furthermore, the cost of a specific technique (precision air seeding), which is not available on every farm but is generally widespread in practice, is not included.

Purchasing costs for Startec are currently 50% above the price for DAP (Keyword 1). The application of the organomineral microgranular fertilizer avoids additional costs for slurry removal (Keyword 5), which more than compensates for the fertilizer’s higher price. As described above, the slurry included in the calculation is especially nutrient-rich, meaning that in practice, the required reduction in the quantity of slurry will tend to be higher, while the average removal costs per m^3^ will not be significantly lower in the region in question. Furthermore, the economic value of the nitrogen, potassium, and magnesium in the slurry is not included in this calculation. However, taking into account the higher crop yields obtained using Startec (Keywords 6, 8–10), the cost–benefit ratio for using microgranular fertilizers (0.028) is lower. This thus points to a more efficient fertilizing system than that calculated for DAP (0.044) on our study site.

## 5. Conclusions

The organomineral microgranular fertilizer under investigation performed better than diammonium phosphate (DAP) in terms of the dry matter yield of maize plants. The use of a mineral microgranular fertilizer resulted in a similar yield to DAP. Compared to the DAP variant, phosphorus balances (nutrient input from fertilizer minus nutrient removal through harvest) were three times lower for the mineral microgranular fertilizer and nine times lower for the organomineral microgranular fertilizer. Both microgranular fertilizers can therefore serve as an adequate alternative in terms of maintaining high yields while maintaining lower—or even null—nutrient surpluses compared to DAP fertilization in maize cultivation on fertile loamy sand sites in central Europe.

Yields from control plots, which had been amended with slurry, did not differ from those from DAP-fertilized plots. The current form of DAP on fertile soils as a standard fertilizer for maize must be critically questioned, both for its effect on nutrient balances and the sustainable management of finite rock-phosphate resources.

Additional studies employing a parallel setup on different soil types, including extensive analyses of soil-phosphorus patterns and the soil microbiome, still need to be carried out. These would help verify the effects of microgranular fertilizer systems under varying conditions in real-world agricultural practice.

## Figures and Tables

**Figure 1 life-14-00081-f001:**
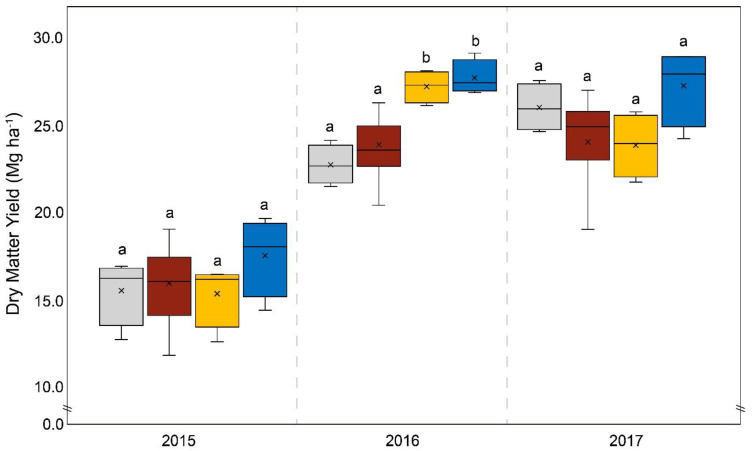
Dry matter yield in Mg obtained per hectare from control plots (grey); diammonium phosphate-fertilized plots (brown); plots treated with WolfNP, a mineral microgranular fertilizer (yellow); and plots treated with Startec, an organomineral microgranular fertilizer (blue) from 2015 to 2017. Different lowercase letters between the plots “a” or “b” indicate a statistically significant difference between variants, *p* < 0.05.

**Figure 2 life-14-00081-f002:**
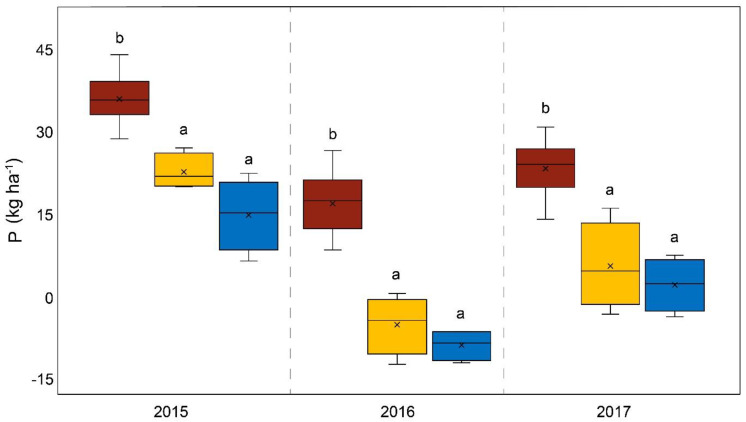
P balance in kg per hectare on diammonium phosphate-fertilized plots (brown); plots treated with WolfNP, a mineral microgranular fertilizer (yellow); plots treated with Startec, an organomineral microgranular fertilizer (blue) from 2015 to 2017. Different lowercase letters between the plots “a” or “b” indicate a statistically significant difference between variants, *p* < 0.05.

**Figure 3 life-14-00081-f003:**
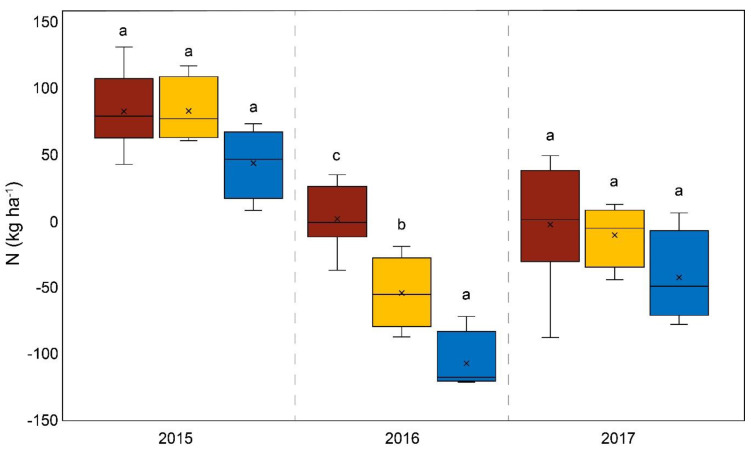
N balance in kg per hectare on diammonium phosphate-fertilized plots (brown); plots treated with WolfNP, a mineral microgranular fertilizer (yellow); and plots treated with Startec, an organomineral microgranular fertilizer (blue) from 2015 to 2017. Different lowercase letters between the plots “a”, “b” or “c” indicate a statistically significant difference between variants, *p* < 0.05.

**Table 1 life-14-00081-t001:** Application rates and nutrient inputs per hectare of fertilizer used.

		Nutrient Type and Input Rate per Hectare							
Fertilizer (Type ^1^)	Application Rate per Hectare in kg	N	P	K	S	Mg	Fe	Mn	Zn
Diammonium phosphate (mineral fertilizer)	100	18	20	-	-	-	-	-	-
Wolf Trax Nu-Trax P+^®^ (mineral microgranular fertilizer)	100	11	0.2	-	8.8	7.5	-	0.4	0.1
Startec (organomineral microgranular fertilizer)	25	1.75	2.4	0.8	1	-	0.125 ^2^	0.125 ^2^	0.375 ^2,3^
Pre-treatment of the whole study site with pig slurry (also used as control)	20,000	172	39	98	48	21.6	-	-	-

^1^ For seed band application. ^2^ EDTA-chelated. ^3^ EDTA-chelated and as oxide. Absent or no data available.

**Table 2 life-14-00081-t002:** Agroeconomic calculation per hectare of variable fertilizer-specific factors.

No.	Keywords	Agroeconomic Calculation of Flexible Fertilizer-Specific Factors per Hectare			
		DAP		Startec	
1	Cost	€40		€60	
2	P balance (excess)	25.5 kg		2.7 kg	
3	kg above permitted P excess *	21.2 kg		0.0 kg	
4	Excess volume of slurry	8.6 m^3^		0.0 m^3^	
5	Additional costs for slurry removal **	€41.11		€0.00	
	Interim result for total costs	€82.11		€60.00	
6	Fresh matter yield	60.99 Mg		69.23 Mg	
7	Loss through ensilage	12%		12%	
8	Silage yield	53.67 Mg	€1878.45	60.92 Mg	€2132.20
9	Remaining profit minus expenses	€1796.34		€2072.20	
10	Surplus compared to DAP fertilization	-		€275.86	
	Cost–benefit ratio	0.044		0.028	

* Permitted P excess based on national German law applicable at the study site. ** Cost of slurry removal based on average data from 14 surveyed farms in northwest Germany.

## Data Availability

The datasets generated and/or analyzed over the course of this study are available from the corresponding author upon reasonable request.

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
