# Peer review of "The Potential of Microgranular Fertilizers to Reduce Nutrient Surpluses When Growing Maize (Zea mays) in Regions with High Livestock Farming Intensity"

_life, 2024, doi:10.3390/life14010081_

Round 1

Reviewer 1 Report (Previous Reviewer 2)

Comments and Suggestions for Authors

Accept in present form

Reviewer 2 Report (Previous Reviewer 1)

Comments and Suggestions for Authors

Dear authors and editor,

The manuscript “The Potential of Microgranular Fertilizers to Reduce Nutrient Surpluses When Growing Maize (Zea mays) in Regions with High Livestock Farming Intensity” has been revised. English was improved, and a new analysis was added to the results. I think the manuscript has reached a quality that makes it suitable for publication.

This manuscript is a resubmission of an earlier submission. The following is a list of the peer review reports and author responses from that submission.

Round 1

Reviewer 1 Report

Comments and Suggestions for Authors

Dear authors and editor,

The manuscript “Microgranular Fertilizers as an option to reduce nutrient surpluses when growing Maize (Zea mays) in regions with high livestock farming intensity” aims to compare two microgranular fertilisers and DAP band fertilisation to assess their impact on dry matter and N and P accumulation. The topic is quite interesting and relevant. Exploring microgranular fertilisers is undoubtedly valuable and aligns well with current research trends.

However, I believe there is an opportunity to enhance the manuscript impact and contribution. While the subject matter is engaging, I noticed that the depth of analyses provided is somewhat limited. The proportion between results and discussion is clearly unbalanced. To truly leverage the potential of your findings and insights, I encourage you to consider repositioning the manuscript as a "Technical Note" rather than a traditional research article. Shifting to the format of a Technical Note would provide several advantages. It would allow you to focus on presenting concise yet impactful results, highlighting the key methodologies employed, and emphasising the practical implications of your work.

Moreover, English writing must be improved: the manuscript is currently hard to read.

 Here are some specific comments:

Line 13: you mention “an organomineral microgranular fertilizer” but you actually analysed two fertilisers

 Line 59 (and elsewhere): I suggest deciding whether to call it maize or corn

Line 65-66: the subject of this sentence is not clear

Line 96: what do you mean about cultivating maize for years? Rotation was not applied?

Why do you use letters indicating significant differences the other way around in the plots? I would expect that higher values are marked with "a".

Comments on the Quality of English Language

There are some areas where the clarity and coherence of the manuscript's language could be improved. Specifically, there are instances where sentence construction and English writing style could be enhanced. Clear and concise language would facilitate understanding your work.

I recommend a thorough review of the manuscript with an emphasis on:

- Sentence structure: Ensure that each sentence has a clear subject, verb, and object. Avoid overly complex sentence structures.

- Clarity: Prioritise clarity in conveying your key findings, methodologies, and implications. Ambiguities in language can confuse readers.

- Proofreading: Pay close attention to grammar, punctuation, and spelling errors.

Reviewer 2 Report

Comments and Suggestions for Authors

An agroeconomic and ecological evaluation of the application of a mineral and an organomineral microgranular fertilizer in contrast to a broad spread mineral band fertilizer in temperate climate regions is intended in this research. The study is of interest, and the subject is up-to-date and suitable, especially for the agricultural field.

Certainly, a more complex approach, in which the aspects of corn production and production quality would have been investigated, together with the results of fertilizations would have offered another perspective and research value. The effects of two different types of microgranular fertilizer (with organic compounds, respectively only mineral), tested in comparison to DAP band fertilization, would be of significant interest in correlation with the most important aspects of corn production and yield quality.

In addition, based only on the analyzes presented in the manuscript (i.e., Figures 1-3, respectively 'Dry matter yield in Mg gained per hectare', 'P balance in kg per hectare', and 'N-balance in kg per hectare'), some approaches would have been better to cover the conclusion below:

Line L334-336: “Both microgranular fertilizers can be considered an adequate alternative to DAP fertilization in maize cultivation on fertile loamy sand sites in central Europe”.

It is probably the same with:

·       "ecological account" (in Abstract, L 12);

·       "ecological potential of the tested fertilizing systems" (L 80-81, at the end of the Introduction, in 'purpose of the investigation');

·       "ecological impact" (in Discussion, L 256-257);

·       "ecology and sustainable management" (in Conclusions (L 339-340).

This, because the ecological evaluation (which would be very complex) is not enough relevant based on the results, and the ecological references do not even appear in the results, but only in the four places above, in the entire manuscript. Probably, reformulating or mitigating the subject at the beginning of the abstract or conclusions could solve these aspects.

A revision of the manuscript could clarify these issues and the authors could provide readers with an easier understanding of their research, results, main findings, and their relevance.

The authors must put themselves in the position of potential readers who are not strictly specialists in the subject of the manuscript or do not clearly understand some notions, terms, or abbreviations. For them, some aspects should be better and more clearly defined, including those that may seem simple (e.g. starting from units of measure or chemical elements, i.e., 'mg' - milligrams, 'Mg' - magnesium; 'Mg' - Mg ha−1, as megagram per hectare for the mass of plant or plant part in the case of yield, see units and abbreviation on https://www.agronomy.org/files/publications/style/chapter-07.pdf; also for 'yield' notion, as e.g. as dry matter 'yield', for the whole plant, per hectare vs. 'yield' for silage or fresh matter yield; the three references to 'Keywords' from L 317-327 which seem unclear, where you refer to column 2 of Table 2 and maybe it would be better to use 'parameter' or 'specific feature' etc.).

 Structure of ms

Please revise the structure of the manuscript. Half of the Discussions chapter actually belongs to the Results chapter! The subchapter/Section "4.2. Economic Consideration of DAP and Organomineral Microgranular Fertilizing Systems" contains the result of your study.

Therefore, you should restructured/reformulate, and the economic issues discussed in connection with appropriate studies.

Statistics

Affirm that differences between fertilizer variants were tested using Student's t-test (L141-142). But in the Figures 1-3 you present comparisons between 4 respectively 3 treatments, which is supposed to be ANOVA (even if they are the same). The Student's t test is used to compare the means of two groups, and there is no need for multiple comparisons because each group has a unique P value, whereas ANOVA is used to compare the means of three or more groups. Or did you use the multiple t test? Not ANOVA with a posthoc?

Anyway, the explanations under these figures must be completed, to understand how you determined the significance of the differences and the significance limits for p-value. Also, explain what exponential function you used to transform the data and ensure the data is normally distributed.

Be careful to avoid some inadvertences and technical editing or spelling mistakes, i.e.:

·       -Check if you have included some data that you refer to (i.e., "crude protein"??? and other, e.g., corncob ratio). See L 136-138: "Shredded material of corncob and remaining plant respectively for each of the four repetitions of a variant was used to determine the content of dry matter, crude protein, phosphorus and potassium.

·       -In the title you use ‘Fertilizers’ and ‘Mayze’ with capital letters for common nouns. Anyway, the journal requires you to capitalize every word in the title, except the connecting ones.

·       ‘yieldcincreasing’ (L290)

Comments on the Quality of English Language

OK